# Identification of SARS-CoV-2 Mpro inhibitors containing P1' 4-fluorobenzothiazole moiety highly active against SARS-CoV-2

Nobuyo Higashi-Kuwata [1], Kohei Tsuji [2], Hironori Hayashi [3], Haydar Bulut [4], Maki Kiso[5], Masaki Imai[5,6], Hiromi Ogata-Aoki [4], Takahiro Ishii [2], Takuya Kobayakawa [2], Kenta Nakano[7], Nobutoki Takamune [8], Naoki Kishimoto [8], Shin-ichiro Hattori [1], Debananda Das [4], Yukari Uemura[9], Yosuke Shimizu [9], Manabu Aoki [4], Kazuya Hasegawa [10], Satoshi Suzuki [11], Akie Nishiyama [11], Junji Saruwatari [12], Yukiko Shimizu [7], Yoshikazu Sukenaga[1], Yuki Takamatsu [1], Kiyoto Tsuchiya [13], Kenji Maeda[1], Kazuhisa Yoshimura[14], Shun Iida [15], Seiya Ozono[15], Tadaki Suzuki [15], Tadashi Okamura [7], Shogo Misumi [8], Yoshihiro Kawaoka [5,6,16], Hirokazu Tamamura [2] & Hiroaki Mitsuya [1,4,17] ✉

COVID-19 caused by SARS-CoV-2 has continually been serious threat to public health worldwide. While a few anti-SARS-CoV-2 therapeutics are currently available, their antiviral potency is not sufficient. Here, we identify two orally available 4-fluoro-benzothiazole-containing small molecules, TKB245 and TKB248, which specifically inhibit the enzymatic activity of main protease (Mpro) of SARS-CoV-2 and significantly more potently block the infectivity and replication of various SARS-CoV-2 strains than nirmatrelvir, molnupiravir, and ensitrelvir in cell-based assays employing various target cells. Both compounds also block the replication of Delta and Omicron variants in human-ACE2-knocked-in mice. Native mass spectrometric analysis reveals that both compounds bind to dimer Mpro, apparently promoting Mpro dimerization. X-ray crystallographic analysis shows that both compounds bind to Mpro's active-site cavity, forming a covalent bond with the catalytic amino acid Cys-145 with the 4-fluorine of the benzothiazole moiety pointed to solvent. The data suggest that TKB245 and TKB248 might serve as potential therapeutics for COVID-19 and shed light upon further optimization to develop more potent and safer anti-SARS-CoV-2 therapeutics.

The novel coronavirus disease 2019 (COVID-19) caused by a positive-strand RNA virus, severe acute respiratory syndrome coronavirus 2 (SARS-CoV-2), has continually been serious threat to public health worldwide[1–3]. Vaccines against SARS-CoV-2 were developed with a stunningly fast speed and showed high efficacy[4] (https://covid19.who.int/). However, the emergence of various SARS-CoV-2 variants[5] with mutations in the spike-encoding region has raised global concerns about the efficacy of the presently available vaccines[6]. At present, only three anti-SARS-CoV-2 therapeutics (remdesivir, nirmatrelvir, and molnupiravir) have been clinically available[7,8] however, the antiviral potency of these three compounds does not seem to be sufficient enough. Indeed, the apparently most potent therapeutic among the

three, nirmatrelvir, reportedly permits the "rebound" of SARS-CoV-2 replication and COVID-19 relapse symptoms and such patients continue to produce contagious virions following the completion of the FDA-recommended 5-day oral administration[9]. Moreover, the emergence of SARS-CoV-2-resistant variants against these first-generation drugs has raised significant concerns[10,11] in part because of the insufficient or only moderate potency of those drugs[12] and in part because of such fast replication speed of SARS-CoV-2, permitting the emergence of drug-resistant variants[13]. The recent record of the high proportion of vaccinated but hospitalized patients suggests not only a need to stay up to date with vaccination and additional booster doses but also for increased use of early outpatient antiviral treatment. Thus, the development of more potent, safer, and more tolerable therapeutics for COVID-19 is desperately required.

Here, based on the antiviral, pharmacologic, and structural features of anti-SARS-CoV-2 $M^{pro}$ inhibitors, GRL1720, GRL0920, and 5h[14–16], we have identified two novel compounds, TKB245 and TKB248, which contain a 4-fluorinated benzothiazole moiety, potently block the infectivity of SARS-CoV-2 through specifically inhibiting the enzymatic activity of $M^{pro}$. Importantly, TKB245 is highly potent with $EC_{50}$ values of ~30 nM as assessed in cell-based assays, exerting significantly more potent activity than the two existing anti-SARS-CoV-2 drugs: molnupiravir[17] and nirmatrelvir[8].

## Results
### Identification of TKB245 with optimized antiviral and pharmacologic features
In the present study, based on the antiviral and structural features of a benzothiazole-containing $M^{pro}$ inhibitor, 5 h[15,18,19], we designed and synthesized a variety of analogs and determined their virological, enzymological, and pharmacological features. Fluorination of certain compounds often enhances their lipophilicity, cell membrane penetration, and oral bioavailability, since the carbon-fluorine bond is more hydrophobic than the carbon-hydrogen bond[19–24]. We thus synthesized a series of fluorinated analogs of 5 h by introducing a fluorine atom(s) on its phenyl ring. We first introduced a fluorine atom to the 4-methoxy-indole ring of 5 h. The first fluorinated 5 h analog, 7-fluoro-4-methoxy-indole-contaning TKB125, had $EC_{50}$ and $EC_{95}$ values similar to that of 5h as determined in cell-based assays employing VeroE6 cells, although its anti-$M^{pro}$ enzymatic inhibition was improved (Table 1). When we added another fluorine atom at the position 4 of the benzothiazole ring of TKB125, generating TKB198, its $EC_{50}$ and $EC_{95}$ values substantially improved by about 10-fold (Table 1). Moreover, when the 7-fluoro-4-methoxy-indole ring at P3 and carbamide at P2 of TKB198 were further replaced with trifluoroacetyl L-α-*tert*-butylglycine and "6,6-dimethyl-3-azabicyclohexane", respectively, the resultant TKB245 achieved highly potent anti-SARS-CoV-2 activity with the $EC_{50}$ and $EC_{95}$ values of 0.03 and 0.53 μM, respectively, and its anti-$M^{pro}$-enzymatic activity was also improved with $IC_{50}$ value of 0.007 μM. We further determined that neither of human cysteine enzymes, cathepsin L and calpain, were significantly affected by TKB245 and TKB248 (Supplementary Fig. 1), which verified the target specificity of both compounds. The pharmacokinetic parameters were also improved: the $T_{1/2}$ of 5 h was as short as 0.27 h, but that of TKB245 became 3.82 h. In an attempt to reduce the potential hydrolytic attack to TKB245, its carbonyl was replaced with thiocarbonyl, generating TKB248. In fact, the $T_{1/2}$ of TKB248 was further elongated to 4.34 h (Table 1), although TKB248's anti-SARS-CoV-2 activity was weakened compared to TKB245's activity. In summary, the antiviral and pharmacologic parameters of TKB245 surpassed the parameters of nirmtrelvir, the first and only $M^{pro}$ inhibitor that has currently been granted emergency use authorization (EUA) by the US FDA for treating COVID-19 (Table 1).

### TKB245 is highly and comparably active against all SARS-CoV-2 variants tested
The continuing emergence of SARS-CoV-2 variants have posed serious concerns. In fact, the efficacy of otherwise highly effective mRNA vaccines has been shown to be significantly blunted and a number of monoclonal antibody-based therapeutics have lost their efficacy. However, the amino acid sequences and structures of various SARS-CoV-2 $M^{pro}$ have been known to be closely related to those from other betacoronaviruses and indeed, various experimental $M^{pro}$ inhibitors examined have been shown to be virtually comparably effective against multiple SARS-CoV-2 variants[15,25]. We, therefore, asked whether TKB245 and TKB248 were also comparably active against a variety of SARS-CoV-2 variants including alpha, beta, gamma, delta, kappa, and omicron species. TKB245 was found to be active against all of such variants examined with $EC_{50}$ values ranging 0.014 to 0.056 μM. TKB248 was also comparably active against all the variants with $EC_{50}$ values ranging 0.070 to 0.430 μM. The order of anti-SARS-CoV-2 potency against variants was: TKB245 » TKB248 > nirmatrelvir (Table 2).

### TKB245 and TKB248 exert potent activity against SARS-CoV-2 in various target cells
We also examined if TKB245 and TKB248 exerted activity against SARS-CoV-2 in various target cells since the antiviral activity of certain antiviral agents significantly vary depending on the target cells employed[8,26]. Since VeroE6 cells are known to abundantly express P-glycoprotein 1 (permeability glycoprotein: P-gp), we determined the anti-SARS-CoV-2 activity in VeroE6 cells in the absence and presence of CP100356, a specific P-gp inhibitor. In the presence of CP100356, three compounds, TKB245, TKB248, and nirmatrelvir, were highly potent with $EC_{50}$ values of 0.001, 0.03, and 0.025 μM, respectively, showing that CP100356 substantially potentiated their antiviral activity by 33-, 3.7-, and 28-fold compared to their activity in the absence of the inhibitor (Table 3). When we determined the anti-SARS-CoV-2 activity in two additional cell lines: (i) HeLa-ACE2-TMPRSS2 cells[27] originally derived from a human cervical cancer cell line, and (ii) A549-ACE2-TMPRSS2 cells derived from a human adenocarcinomic alveolar-basal epithelial line, both of which were rendered to express human ACE2 and TMPRSS2, TKB245 exerted potent activity against SARS-CoV-2$^{TWK521}$ with $EC_{50}$ values of 0.021 and 0.0027 μM, respectively.

We also evaluated the anti-SARS-CoV-2 activity of ensitrelvir[28], another $M^{pro}$ inhibitor under clinical trial, and molnupiralvir[17], a synthetic nucleoside derivative presently clinical available under EUA, using VeroE6 cells and HeLa-ACE2-TMPRSS2 cells, and the $EC_{50}$ values were determined in side-by-side antiviral assays. Among the five compounds, TKB245 exerted the most potent activity in any of the target cells employed (Table 3).

### Intracellular concentrations and PKs of TKB245, TKB248 and nirmatrelvir
Once a drug enters into systemic circulation, the drug should eventually enter the cells or tissues to pharmaceutically exert its activity. Thus, we determined the intracellular concentrations of TKB245, TKB248, and nirmatrelvir in VeroE6 cells and HeLa-ACE2-TMPRSS2 cells after exposing them to each of the three $M^{pro}$ inhibitors using LC/MS. When exposed to nirmatrelvir, substantial amounts of the compound were seen in both cells preincubated at 50 and 100 μM (Fig. 1). However, the amounts of TKB245 at 50 μM were significantly greater by 39- and 2.3-fold at 100 μM, and by 19.5- and 2.3-fold as compared to those with nirmatrelvir in VeroE6 cells and HeLa-ACE2-TMPRSS2 cells, respectively. The concentrations of TKB248 were furthermore greater at 50 μM by 4800-fold and 318-fold, and at 100 μM by 2289-fold and 165-fold in VeroE6 cells and HeLa-ACE2-TMPRSS2 cells, respectively, as compared to nirmatrelvir (Fig. 1). The replacement of the carbonyl of TKB245 with

**Table 1 | In vitro and in vivo parameters of M^pro inhibitors examined in the present study**

| Compound | Structure | SARS- CoV2 RNA VeroE6 cell-based assay (µM)** | | SARS- CoV2 M^pro enzyme assay (µM)* | | T1/2# (h) | Cmax # (ng/mL) | Oral F# (%) |
|---|---|---|---|---|---|---|---|---|
| | | EC_{50} | EC_{95} | IC_{50} | IC_{95} | | | |
| 5h M.W. 575.6840 | | 2.60 ± 0.50 | 9.47 ± 0.33 | 0.13 ± 0.11 | 1.5 ± 0.6 | 0.27§ | n.d. | n.d. |
| TKB125 M.W. 593.6744 | | 1.82 ± 0.96 | 8.39 ± 0.39 | 0.034 ± 0.006 | 1.02 ± 0.40 | 0.76 § | n.d. | n.d. |
| TKB198 M.W. 611.6648 | | 0.27 ± 0.11 | 0.83 ± 0.03 | 0.023 ± 0.013 | 0.75 ± 0.27 | 0.83 § | 85 | 1.98§ |
| TKB245 M.W. 653.6936 | | 0.03 ± 0.02 | 0.53 ± 0.33 | 0.007 ± 0.002 | 0.14 ± 0.07 | 3.82¶ | 1,901¶ | 48¶ |
| TKB248 M.W. 669.7546 | | 0.22 ± 0.08 | 0.87 ± 0.04 | 0.074 ± 0.034 | 6.00 ± 5.29 | 4.34¶ | 1,925¶ | 72¶ |
| Nirmatrelvir M.W. 499.5352 | | 0.94 ± 0.21 | 7.81 ± 1.55 | 0.013 ± 0.004 | 0.77 ± 0.25 | 1.03¶ | 1,157¶ | 56¶ |

Fifty % inhibitory concentration (IC_{50}) and 50% effective concentration (EC_{50}) values were calculated as previously published[14]. Data from three-four independent assays are shown as arithmetic means ± 1 S.D.

# Pharmacokinetic parameters were calculated using plasma concentration–time data and are shown as mean values.

§N = 2 to 3 male or female ICR mice.

¶N = 3 male PXB-mouse 10 mg/kg i.v. and p.o. (see Supplementary Fig. 2 for details). F is defined as the dose-normalized AUC after oral administration divided by the dose-normalized AUC after intravenous administration.

*Inhibition of SARS-CoV2 M^pro enzyme activity by compounds was measured with fluorescence resonance energy transfer (FRET) assay system.

**Cell-based anti-SARS-CoV2 activity by compounds was determined with RT-qPCR assays of viral RNA from SARS-CoV2-exposed VeroE6 cells. Source data are provided as a Source Data file.

**Table 2 | TKB245 exerts potent activity against various SARS-CoV-2 variants**

| | Against SARS-CoV-2 variants EC$_{50}$ (µM) | | | | | | | | | |
|---|---|---|---|---|---|---|---|---|---|---|
| Compound | WK521 | QHN001/α | TY8-612/β | Y7-501/γ | K1734/δ | 5356/k | KTKYX00012/o BA.1 | TKYVS2037/o BA.2 | TKYTS14631/o BA.5 | UT-NCD1757-1N/o BA.2.75 |
| TKB245 | 0.03 ± 0.02 | 0.042 ± 0.042 | 0.028 ± 0.009 | 0.056 ± 0.045 | 0.030 ± 0.003 | 0.020 ± 0.002 | 0.014 ± 0.001 | 0.045 ± 0.014 | 0.019 ± 0.005 | 0.049 ± 0.005 |
| TKB248 | 0.22 ± 0.08 | 0.300 ± 0.001 | 0.287 ± 0.005 | 0.228 ± 0.061 | 0.273 ± 0.036 | 0.180 ± 0.035 | 0.282 ± 0.043 | 0.210 ± 0.087 | 0.070 ± 0.008 | 0.430 ± 0.091 |
| Nirmatrelvir | 0.94 ± 0.21 | 1.980 ± 0.559 | 1.498 ± 0.240 | 2.494 ± 0.292 | 1.499 ± 0.078 | 1.658 ± 0.606 | 1.047 ± 0.061 | 1.240 ± 0.125 | 1.080 ± 0.150 | 0.967 ± 0.135 |

Anti-SARS-CoV2 activity by compounds was evaluated with RT-qPCR of viral RNA from culture medium of SARS-CoV-2-exposed VeroE6 cells. Each EC$_{50}$ value was calculated as previously published[15]. Data from three-four independent assays are shown as arithmetic means ± 1 S.D. Source data are provided as a Source Data file.

**Table 3 | TKB245 demonstrates potent SARS-CoV-2$^{WK521}$ antiviral activity and favorable cytotoxicity profiles**

| Cell line | TKB245 (µM) | | TKB248 (µM) | | Nirmatrelvir (µM) | | Ensitrelvir fumarate (µM) | | Molnupiravir (µM) | |
|---|---|---|---|---|---|---|---|---|---|---|
| | EC$_{50}$ | CC$_{50}$ | EC$_{50}$ | CC$_{50}$ | EC$_{50}$ | CC$_{50}$ | EC$_{50}$ | CC$_{50}$ | EC$_{50}$ | CC$_{50}$ |
| VeroE6 | 0.03 ± 0.02 | >100 | 0.22 ± 0.08 | >100 | 1.02 ± 0.34 | >100 | 0.11 ± 0.05 | >100 | 0.35 ± 0.06 | >100 |
| VeroE6 + 2 µM CP100356 | 0.001 ± 0.001 | >100 | 0.03 ± 0.03 | >100 | 0.025 ± 0.002 | >100 | - | - | - | - |
| HeLa-ACE2-TMPRSS2 | 0.021 ± 0.008 | >100 | 0.16 ± 0.075 | >100 | 0.24 ± 0.05 | >100 | 0.20 ± 0.03 | >100 | 0.32 ± 0.07 | >100 |
| A549-ACE2-TMPRSS2 | 0.0027 ± 0.0002 | >100 | 0.199 ± 0.005 | >100 | 0.017 ± 0.003 | >100 | - | - | - | - |
| HepG2 | - | >200 | - | >200 | - | >200 | - | - | - | - |

TKB245 inhibited SARS-CoV-2$^{WK521}$ replication in VeroE6, HeLa-ACE2-TMPRSS2, and A549-ACE2-TMPRSS2 cells. A P-glycoprotein inhibitor, CP-100356 (efflux inhibitor, EI, 2 µM), was added to inhibit the P-glycoprotein–mediated efflux of TKB245 in VeroE6 cells. Cytotoxicity of TKB245 was evaluated in noninfected cells and was determined with a water-soluble MTT assay. Data from three-four independent assays are shown as arithmetic means (µM) ± 1 S.D. Source data are provided as a Source Data file.

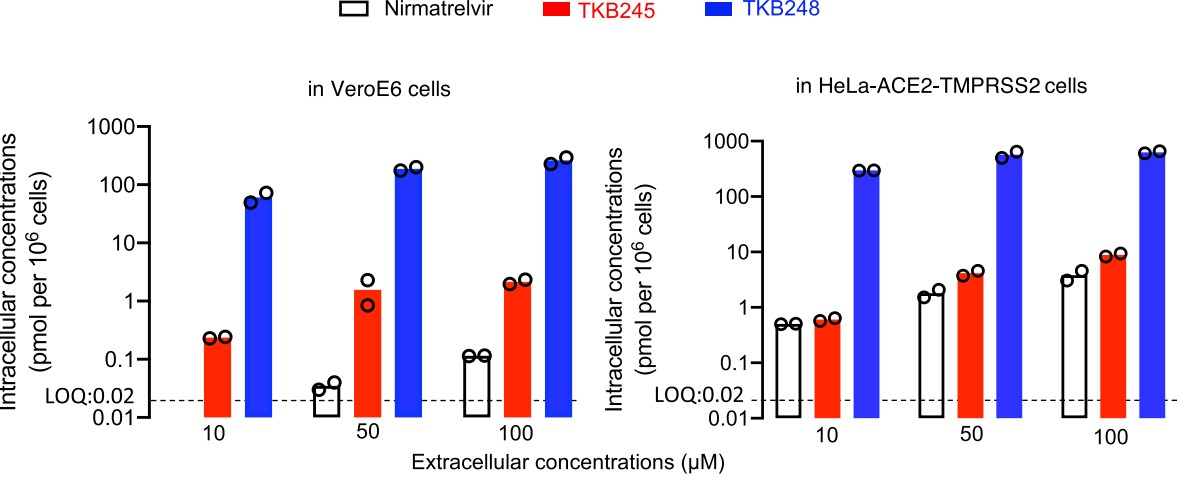

**Fig. 1 | Intracellular concentrations of TKB245 and TKB248.** VeroE6 or HeLa-ACE2-TMPRSS2 cells were incubated with 10, 50, and 100 μM of each compound for 6 hours, vigorously washed with PBS, and intracellular concentrations of each compound were determined using LC/MS. Bars indicate arithmetic means ($n = 2$). LOQ: Limit of Quantitation. Source data are provided as a Source Data file.

thiocarbonyl, generating TKB248, obviously increased TKB248's intracellular concentrations in both cell lines (Fig. 1); however, the mechanism of the increase remains to be elucidated.

### Effects of TKB245 and TKB248 on the replication of SARS-CoV-2$_{NC928-2N}$$^{Omicron/BA.1}$, SARS-CoV-2$_{UW5250}$$^{Delta}$, and SARS-CoV-2$_{NCD1288}$$^{Omicron/BA.2}$ variants in human ACE2-knocked-in mice

In order to examine the in vivo efficacy of TKB245 and TKB248 in small animals, we first determined their pharmacokinetics using human liver-chimeric (PXB) mice, which are thought to be of utility in addressing early safety assessment and possible characterization of drug metabolism that occurs in humans[29]. When TKB245, TKB248, and nirmatrelvir were orally (intra-gastrically) administered (10 mg/kg, $n = 3$ for each compound) and plasma concentrations of each compound were determined, TKB248's plasma concentrations persisted the longest, followed by TKB245, and then nirmatrelvir concentrations. TKB245's $t_{1/2}$ values were 1.25 and 3.82 hours when administered intravenously and orally, respectively. The oral bioavailability values of TKB245 and TKB248 were 48% and 72%, respectively (Supplementary Fig. 2).

Based on the pharmacokinetics determined as above, we assessed the therapeutic efficacy of TKB245 and TKB248 in human ACE2-knocked-in mice following SARS-CoV-2 exposure. Five mice per group were intranasally inoculated with SARS-CoV-2$_{NC928-2N}$$^{Omicron/BA.1}$ ($5 \times 10^5$ PFU) or SARS-CoV-2$_{UW-5250}$$^{Delta}$ ($5 \times 10^5$ PFU), then 2 hours later intraperitoneally administered with TKB245 (100 mg/kg) or vehicle (methylcellulose as control), and were euthanized at 48 and 72 h post-inoculation. Lungs were collected, homogenized, and infectious viral titers in those tissue homogenates were determined by using VeroE6$^{TMPRSS2}$ cells[15]. TKB245 considerably reduced titers of both variants in lung (Fig. 2a). Although the levels of reduction by TKB245 appeared to be greater in mice exposed to the Omicron variant than in those exposed to the Delta variant, the virus titers in mice receiving the vehicle were lesser for the Omicron variant, implying that TKB245 apparently exerted more potently. TKB248 also exerted comparable efficacy against both Omicron BA.1 and BA.2 variants (Fig. 2b). When the body weights and survival rates of the mice receiving TKB245, TKB248 or the vehicle-only were determined pre-infection and on day 3 post-infection, no significant changes were recorded regardless of the exposure to either of the variants (Supplementary Fig. 3), suggesting that no acute toxicity of TKB245 and TKB248 was present. No significant changes in the lung images obtained with the micro-computed tomography (micro-CT) were recorded (Supplementary

Fig. 4a) when tested pre-infection and on day 3 post-infection. Moreover, histopathological examination of lung tissues revealed overall no significant differences in the distribution or replication levels of either of the two variants (Supplementary Fig. 4b). Nevertheless, in comparison of viral titers between vehicle only (no-drug control) and TAK245-/TKB248-treated mice using repeated ANOVA with days and mutant strains as fixed effects and individuals as a random effect, TKB245 showed significantly lower virus titer compared to the no-drug control [-1.34 (95%CI: -1.85, -0.82)]. Similarly, TKB248 showed significantly lower virus titers compared to no-drug controls [-1.01 (95% CI: -1.41, -0.60)].

### TKB245 and nirmatrelvir bind to M$^{pro}$ and promote M$^{pro}$ dimerization

The modes of the binding of TKB245, TKB248, and nirmatrelvir to SARS-CoV-2 M$^{pro}$ was examined using native mass spectrometric analysis. The data showed that in the absence of inhibitors, the monomer-dimer equilibrium of SARS-CoV-2 M$^{pro}$ shifted toward more abundant dimers and fewer monomers (Fig. 3a, Supplementary Fig. 5). As shown in Fig. 3a, the presence of 15 μM of TKB245 and nirmatrelvir shifted the equilibrium more toward a predominant abundance of dimers bound by mostly two inhibitor molecules and left no significant amounts of monomers, suggesting that these two compounds inhibited the enzymatic activity in a manner, in which the two inhibitor molecules bind to M$^{pro}$ and promote and stabilize the dimerization. On the other hand, unlike in the case of TKB245 or nirmatrelvir, TKB248 at 15 μM did not significantly change the amounts of monomers or dimers although one or two TKB248 molecules were found to be bound to the enzyme. However, the presence of 50 μM TKB248 significantly increased the amount of M$^{pro}$ bound by two TKB248 molecules, followed by that of M$^{pro}$ bound by one molecule. (Supplementary Fig. 6), suggesting that all three M$^{pro}$ inhibitors similarly promote the dimerization of two protomers. Considering that the M$^{pro}$ enzymatic inhibitory activity of TKB248 (IC$_{50}$ = 0.074 μM) was much less than that of TKB245 (IC$_{50}$ = 0.007 μM)(Table 1), the lesser binding affinity of TKB248 resulted in its lesser anti-SARS-CoV-2 activity. Since Ser-1 is thought to be critical for M$^{pro}$ dimerization, we added an amino acid (glycine) to Ser-1. The equilibrium of the glycine-added M$^{pro}$ significantly shifted to its monomer form and substantially lesser levels of the three M$^{pro}$ inhibitors bound to M$^{pro}$ (Fig. 3b, Supplementary Figure 5), strongly suggesting that the Ser-1 plays a substantial role in the initiation and/or progression of the dimerization process[30,31].

**a**

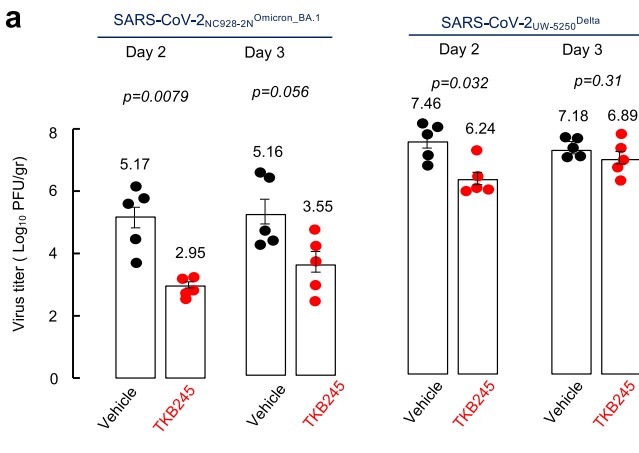

**b**

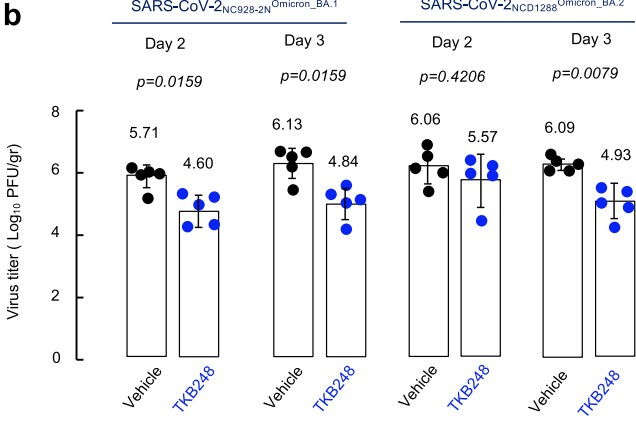

**Fig. 2 | In vivo efficacy of TKB245 and TKB248 against SARS-CoV-2NC928-2NOmicron_BA.1, SARS-CoV-2UW-5250Delta and SARS-CoV-2NCD1288Omicron_BA.2-infected hACE2-knocked-in mice. a** Five hACE2KI mice per group were challenged intranasally with SARS-CoV-2NC928-2NOmicron (5 × 10^5 PFU) or SARS-CoV-2UW-5250Delta (5 × 10^5 PFU). Two hours later, animals were intraperitoneally administered 100 mg/kg BID TKB245 or vehicle (placebo). Animals were euthanized on 2- and 3-days post infection and lungs collected for determination of virus titers using VeroE6TMPRSS2 cells. Bars indicate mean values and error bars represent standard deviations. All *p* values are calculated using the exact Wilcoxon rank-sum test with two-sided, and no multiple adjustment was made. Source data are provided as a Source Data file. **b** Five hACE2K mice per group were challenged intranasally with SARS-CoV-2NC928-2NOmicron_BA.1 (5 × 10^5 PFU) or SARS-CoV-2NCD1288Omicron_BA.2 (1 × 10^5 PFU). Two hours later, animals were intraperitoneally administered 120 mg/kg BID TKB248 or vehicle (placebo). Animals were euthanized on 2- and 3-days post infection and lungs collected for determination of virus titers using VeroE6TMPRSS2 cells. Bars indicate mean values and error bars represent standard deviations. All p values are calculated using the exact Wilcoxon rank-sum test with two-sided, and no multiple adjustment was made. Source data are provided as a Source Data file.

**Structural analysis of TKB245 and TKB248 complexed with M^pro**

Using the X-ray structures, we also examined the binding mode of TKB245 and TKB248 complexed with M^pro (Fig. 4a). Both compounds have been built on dimethyl-bicyclo[3.1.0]-proline moiety in the center, γ-lactam as the P1 moiety, and 4-fluorobenzothiazole as the P1′ moiety. The only difference in chemical composition is the substitution of sulfur (TKB248) with oxygen (TKB245) at the center part (Table 1). Structures of both inhibitors exhibit identical binding mode within the binding pocket (Fig. 4b). Moreover, we observed covalent bond formation with Cys145 in both inhibitors (Fig. 4c, 4f, and 4g). Inside the S1 subsite, the oxygen of the γ-lactam moiety forms a strong hydrogen bond with the side-chain of His-163 (Fig. 4c). In addition, the

nitrogen atom of the γ-lactam moiety forms hydrogen bond with the carboxylate side chain of Glu-166 (Fig. 4c) and with the main-chain oxygen of Phe-140 located at a distance of 3.5 Å (Fig. 4c). Additional hydrogen bonds are formed between the amide group from the center moiety and the main chain oxygen of His-164 (Fig. 4c). The tri-fluoroacetamide group protrudes into the S3 sub-pocket, where it forms favorable halogen interactions with surrounding residues (Fig. 4d). In the center part, dimethyl-bicyclo[3.1.0]-proline moiety effectively fills the hydrophobic S2 pocket formed by Met49, Met165, and His41, effectively replacing the function of hydrophobic Leu (P2) of the virally-coded polyprotein substrate. The S1′ subsite is fully occupied with the 4-fluorobenzothiazole ring; however, the 4-positioned fluorine atom does not form direct contact with any binding pocket residues, instead it points out of the binding pocket (Fig. 4f). The nitrile warhead of nirmatrelvir forms a covalent thioimidate group and its nitrogen atom engages three hydrogen bonds with the oxyanion hole residues with the pattern similar to the cases of TKB245 and TKB248 (Fig. 4e). In the structures of TKB245 and TKB248, we observed covalent bond formation between Cys145 and 4-fluorobenzothioazoly ketone, resulting in a hemithioketal group, with an oxygen atom directed towards the oxyanion hole formed with amino acids of Cys145, Ser144, and Gly143. (Fig. 4f, g). Omitted electron densities (2Fo − Fc) of TKB245 and TKB248 including C145, contoured at the 1σ level are shown inside the binding groove (Supplementary Fig. 7a, b). Of note, nirmatrelvir lacks any P1′ moiety, which leaves the S1′subpocket empty (Fig. 4h). (See crystallography parameters in Supplementary Table 1)

Taken together, the structural data strongly suggest that the 4-fluorobenzothiazole moiety plays a crucial role in the potent activity of TKB245 and TKB248 against SARS-CoV-2 strains.

## Discussion

The two antiviral therapeutics, nirmatrelvir, and molnupiravir, which are clinically available under the emergency use authorization by FDA as of September 2022, are associated with some adverse effects. Side effects of nilmatrelvir include impaired sense of taste, diarrhea, high blood pressure, and muscle aches and is not recommended for people who have severe kidney or liver impairment. Administration of nirmatrelvir with ritonavir can also cause various uncontrollable drug-drug interactions. Adverse effects of molnupiravir includes diarrhea, nausea, and dizziness and is not recommended for use during pregnancy because of potential fetal harm seen in animals. In addition, the continuing emergence of naturally-occurring SARS-CoV-2 variants exemplifies the capability of SARS-CoV-2 to mutate, signifies formidable possibility of the current pandemic to be persistent in the future, and strongly suggests that drug-resistant SARS-CoV-2 variants would make the antiviral treatment less efficacious in the near future. In addition, it has been shown that microorganisms such as viruses and bacteria effectively acquire drug-resistance if they continue to replicate in the presence of insufficient antimicrobial pressure[12]. Moreover, there have been reports that virologic rebound after nirmatrelvir-ritonavir therapy for individuals with early stage COVID-19 infection is associated with high viral load and culturable SARS-CoV-2[32], which might be related to insufficient SARS-CoV-2 suppression with nirmatrelvir. Thus, for more effective response to the present COVID-19 pandemic, more potent and more tolerable antiviral therapeutics are urgently required for treating COVID-19.

Fluorination often increases metabolic stability, delays inactivation of drugs, and elongates dosage periods because C-F bond is highly stable[20,21,29,33]. Fluorination also increases lipophilicity due to its greater hydrophobicity than C-H bond, often increasing cell membrane penetration. Thus, starting from our observation on the previously published coronavirus M^pro inhibitor, 5 h[15,18,19], we designed and synthesized ~100 different 5h analogs, and identified TKB245 and TKB248 as potent SARS-CoV-2's M^pro inhibitors that highly specifically bind to

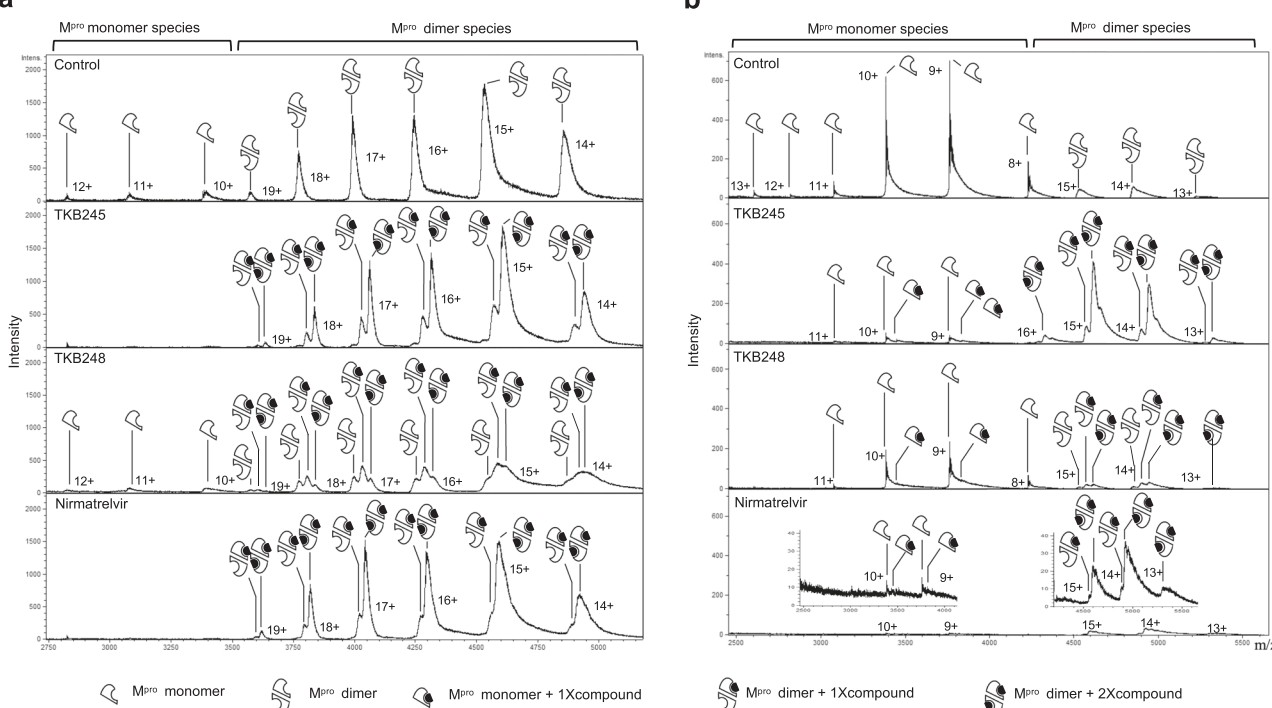

**Fig. 3 | Native mass spectrometric analysis of SARS-CoV-2 M^pro-inhibitor interaction. a** Authentic M^pro (7.5 μM) was treated with 15 μM of each indicated compound. Relative native mass spectra of the M^pro with or without TKB245, TKB248, or nirmatrelvir are shown. Charge states 10⁺, 11⁺ and 12⁺ are annotated to mass spectra corresponding to M^pro monomer species and charge states 14⁺, 15⁺, 16⁺, 17⁺, 18⁺ and 19⁺ are annotated to mass spectra corresponding to M^pro dimer species. **b** Glycine-added M^pro (10 μM) was treated with 50 μM of each indicated compound. Relative native mass spectra of the M^pro with or without TKB245, TKB248, or nirmatrelvir are shown. Charge states 8⁺, 9⁺, 10⁺, 11⁺, 12⁺, and 13⁺ are annotated to mass spectra corresponding to M^pro monomer species and charge states 13⁺, 14⁺, 15⁺, and 16⁺ are annotated to mass spectra corresponding to M^pro dimer species.

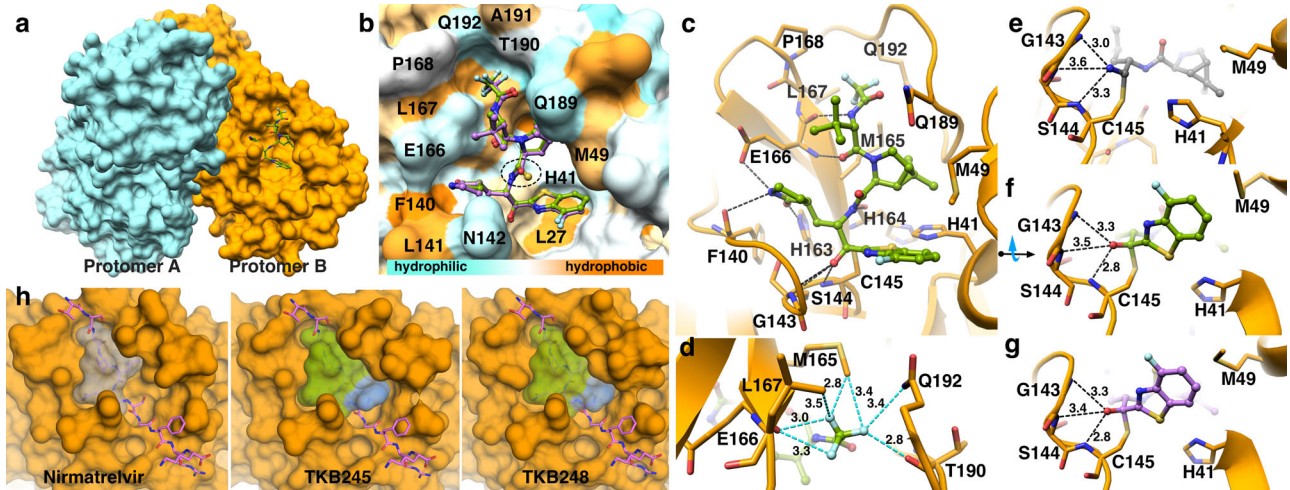

**Fig. 4 | Co-crystal structures of TKB245 and TKB248 with SARS-CoV-2 M^pro.**
**a** Overview of M^pro dimer in complex with TKB245. Molecular surface of protomer A colored in cyan and protomer B in orange. **b** Superimposition of TKB245 in green onto TKB248 in purple exhibits identical binding mode. M^pro binding pocket is colored according to hydrophobicity scale. Hydrophobicity of the binding pocket is represented by the intensities of orange color such as hydrophobic residues: Leu-27 and Phe-140 as shown in orange. Polar or charged residues such as Glu-166, Gln-189 are shown in light blue. **c** Binding mode and hydrogen bond network in M^pro complexed with TKB245. Cartoon representation of the crystal structure of M^pro is shown in orange complexed with TKB245 (green stick). Hydrogen bonds are indicated as black dashed lines. **d** Top view focused on trifluoromethyl (P4) group with potential fluorine-based interaction: Less than 4 Å are indicated as cyan dashed lines. Three fluorine atoms engaged in multi-directional halogen bond interactions with surrounding residues consist of Leu-167, Pro-168, Gln-192, and Met-165. **e–g** Comparison of nirmatrelvir (PBD ID: 7RFW) *vs* TKB245 (PBD ID: 8DOX) and TKB248 (PBD ID: 8DPR) inside the S1′ subsite including oxyanion hole interactions. While all inhibitors covalently bind to the catalytic residue Cys-145, the 4-fluorobenzothiazole ring with the fluorine atom points to solvent. **h** Side-by-side comparison of nirmatrelvir *vs* TKB-245 and TKB-248 (as shown in transparent surface) onto the polyprotein substrate. Blue color indicates the 4-fluorobenzothiazole ring of TKB245 and TKB248 that effectively fill the S1′subsite compared to nirmatrelvir.

SARS-CoV-2's M^pro, potently block the infectivity and replication of a variety of SARS-CoV-2 variants as examined with cell-based assays, and have favorable pharmacokinetic features (Table 1).

Unlike in the case of Syrian Hamsters infected with SARS-CoV-2 that undergo severe lung damages but survive such damages[34,35], the human-ACE-2-knocked-in mice used in the present study did not undergo significant inflammations or consolidations in the lung when examined with micro-CT or histochemistry. Thus micro-CT images or histopathology findings could not be utilized to evaluate possible effects of antiviral agents on the virus-inflicted damages. However, when viral titers in the homogenized lung tissues were determined, significant to substantial differences were seen between mice receiving the vehicle only and those receiving TKB245 or TKB248. However, of note, in order to evaluate pharmacological and antiviral efficacy of test compounds in the early phase of development, the use of mice that weigh 20 to 35 grams each is of help since a relatively small amount of agent suffices the experimental research, compared to Syrian hamsters or rats, that weigh ~150 to ~260 g that demands relatively large amounts of test compounds.

The present native mass spectrometric analysis revealed that the addition of TKB245 and nirmatrelvir substantially shifted the equilibrium toward a greater abundance of dimers, predominantly bound by two inhibitor molecules, suggesting that these two inhibitors inhibit the enzymatic activity in a manner in which these inhibitors expedite dimerization and stabilize the dimerization of monomers to the extent so that M^pro inhibitors occupy the enzymatic active site hydrophobic cavity and prevent the entry of SARS-CoV-2 immature polyproteins, which are otherwise processed by M^pro (Fig. 4h). The promoted protease dimerization triggered by its inhibitors has also been seen in the interactions of HIV-1 protease and its inhibitors, in which in the presence of HIV-1 protease inhibitors (such as saquinavir and nelfinavir) the amounts of HIV-1 protease monomers are reduced to form dimers bound to the inhibitors[36], affirming similar enzyme-specific nature of inhibition by HIV-1 protease inhibitors and M^pro inhibitors.

In the present study, we found that TKB245 exerts more potent activity in inhibiting M^pro's enzymatic activity than nirmatrelvir. One of the significant structural differences is that TKB245 carries ketone (-CO) but nirmatrelvir has nitrile (-CN) as a warhead, which might be associated with the difference of their enzymatic inhibition activity. The exact comparative study of the two warheads remains to be conducted.

Regardless of the mechanism(s), the results reported here demonstrate that a simple fluorination of the benzothiazole can convert an inhibitor that has no fluorine atom in the benzothiazole (i.e., TKB125 in Table 1) to a compound with the capacity to potently inhibit the replication and cytopathic effect of SARS-CoV-2 (i.e., TKB198), resulting in the identification of TKB245. Notably, TKB245 exerts highly more potent activity compared to the two existing anti-SARS-CoV-2 drugs (molnupiravir and nirmatrelvir) and an anti-SARS-CoV-2 agent under development, ensitrelvir. The present results might have implications in the development of new strategies for the pharmacologic intervention against the pathogenic coronaviruses including SARS-CoV-2.

## Methods
### Cells, viruses, and antiviral compounds
VeroE6 cells were obtained from the American Type Culture Collection (ATCC) (CRL-1586) (Manassas, VA) and were maintained in Dulbecco's modified Eagle's medium (d-MEM) supplemented with 10% fetal bovine serum (FCS), 100 μg/ml of penicillin, and 100 μg/ml of streptomycin. HeLa-ACE2-TMPRSS2 cells were obtained from the Japanese Collection of Research Bioresources (JCRB) Cell Bank (JCRB1835, Osaka, Japan) and were maintained the same conditioned medium as that VeroE6 cell line except for G418 (0.5 mg/mL) addition. A549-hACE2-TMPRSS2 cells were purchased from Invivo Gen (San Diego, CA,

USA) (a549-hace2tpsa). and were cultured in growth medium in 0.5 μg/ml of Puromycin and 300 μg/ml of Hygromycein. SARS-CoV-2 strain JPN/TY/WK-521 (SARS-CoV-2^WK-521) was obtained from the National Institute of Infectious Diseases (Tokyo, Japan). Five clinically isolated SARS-CoV-2 mutant strains were used in the current study: a B.1.1.7 (alpha) strain [hCoV-19/Japan/QHN001/2020 (SARS-CoV-2^QHN001, GISAID Accession ID; EPI_ISL_804007)], a B.1.351 (beta) strain [hCoV-19/Japan/TY8-612-P0/2021 (SARS-CoV-2^TY8-612)] and a P.1 (gamma) strain [hCoV-19/Japan/TY7-501-P0/2021 (SARS-CoV-2^TY7-501)] were obtained from National Institute of Infectious Diseases, Tokyo, Japan. A B.1.617.2 (delta) strain [hCoV-19/Japan/TKYK01734/2021 (SARS-CoV-2^K1734, GISAID Accession ID; EPI_ISL_2080609)], a B.1.617.1 (kappa) strain [TKYTK5356_2021 (SARS-CoV-2^5356, DDBJ Accession ID; LC633761)], a BA.1.1.529 (omicron BA.1) strain [hCoV-19/Japan/TKYX00012/2021 (SARS-CoV-2^00012 GISAID Accession ID; EPI_ISL_8559478)], a BA.1.1.529 (omicron BA.2) strain [hCoV-19/Japan/TKYS02037/2022 (SARS-CoV-2^02037, GISAID Accession ID; EPI_ISL_9397331)], and a BA.5 (omicron BA.5) strain [hCoV-19/Japan/TKY TKYTS14631/2022 (SARS-CoV-2^14631, GISAID Accession ID; EPI_ISL_12812500.1)] were provided from Tokyo Metropolitan Institute of public Health, Tokyo, Japan. A B.1.617.2 (delta) strain [hCoV-19/USA/WI-UW-5250/2021 (SARS-CoV-2_UW-5250^Delta, GISAID Accession ID; pending)], a BA.1.18 (omicron BA.1) strain [hCoV-19/Japan/NC928-2N/2021 (SARS-CoV-2_NC928-2N^Omicron_BA.1, GISAID Accession ID; EPI_ISL_7507055)], a BA.2.10 (omicron BA.2) strain [hCoV-19/Japan/UT-NCD1288-2N/2022 (SARS-CoV-2_NCD1288^Omicron_BA.2, GISAID Accession ID; EPI_ISL_9595604)], and a BA.2.75 strain [hCoV-19/Japan/UT-NCD1757-1N/2022 (SARS-CoV-2^UT-NCD1757-1N/o, GISAID Accession ID; EPI_ISL_14321746)] were obtained from the University of Tokyo, Tokyo, Japan. All variants used in this study was listed in Supplementary Table 2. The antiviral agents nirmatrelvir (synonymus:PF-07321332)[8], molnupiravir[17], and ensitrelvir[28] were purchased (MedChemExpress, Mon-mouth Junction, NJ). 5 h was synthesized by A. K.Ghosh. TKB125, TKB198, TKB245, and TKB248 were newly designed and synthesized and their purity was > 95% or 99% as assessed with high-performance liquid chromatography (HPLC). The methods of synthesis of TKB125, and TKB198 were published by Tamamura H et al.[37] and the general methods for synthesis and characterization of all new compounds are shown in Supplementary Methods. Each compound was dissolved in DMSO at 20 mM as stock solutions.

### Antibodies used and validation
For immunocytostaining, COVID-19 convalescent plasma-derived IgG (ConvIgG) was used as a primary antibody (1/500 dilution)(IgG was purified at National Center for Global Health and Medicine), while Alexa Fluor® 488 AffiniPure Fab Fragment Goat Anti-Human IgG (H + L) was used for a secondary antibody (1/200 dilution)(Jackson ImmunoResearch, 109-547-003). A rabbit monoclonal antibody that detects SARS-CoV-2 nucleocapsid protein (1:1,000 dilution, catalog number 40143-R001, Sino Biological, Beijing, China) was used for immunohistochemistry of histopathological examination (Supplimentary Fig. 4b). SARS-CoV-2 infection and IgG amounts were determined with RNA-qPCR and ELISA, respectively. ConvIgG was validated using immunostaining of SARS-CoV-2-infected and -uninfected VeroE6 cells and the data obtained were confirmed to be free from nonspecific detection. The rabbit monoclonal antibody was validated using immunohistostaining of SARS-CoV-2-infected and -uninfected lung of hACE2 knock-in mice and the data obtained were confirmed to be free from nonspecific detection.

### SARS-CoV-2 M^pro/3CL^pro and human cysteine protease enzyme assay
The 3CLpro (SARS-CoV-2) assay kit (BPS Bioscience, San Diego CA, cat. no. 78042-2) is designed to measure 3CLpro activity and identify inhibitors of this enzyme, while human cathepsin L inhibitor screening kit (abcam, cat. Cambridge, UK, No. ab197012) and human calpain

activity assay kit (AnaSpec. Inc. Fremont CA, cat. No. AS-72149) are designed to measure human cysteine protease activity and identify inhibitors of these enzymes. These assays were performed in a 96-well plate using a fluorogenic substrate. Briefly, a solution of each enzyme was prepared according to the manufacturer's protocol in assay buffer. Separately, solutions of test compounds necessary to generate a seven-point dose response curve were prepared in half-log serial dilution. Test compounds were added to the plate, and the mixture was preincubated for 30 min at room temperature. A blank well (no enzyme) was included to assess the background signal, while the known inhibitors GC376, FF-FMK and B27-WT were used as positive controls for SARS-CoV-2 M$^{pro}$/3CL$^{pro}$, cathepsin L, and calpain, respectively. The plates of SARS-CoV-2 M$^{pro}$, cathepsin L, and calpain were incubated with each fluorogenic substrate for 4, 0.3, and 1 h, respectively, at room temperature. Then fluorescence intensity was measured in a Cytation 5 cell imaging multi-mode reader (BioTek, Winooski, VT, USA)(excitation/emission: 360/460 nm, excitation/emission: 400/505 nm, and excitation/emission: 490/520 nm), respectively. End point fluorescence intensities were measured, and the blank value were subtracted from all values.

### Antiviral activity and cytotoxicity assays

For antiviral assay, cells were seeded in a 96-well plate ($2 \times 10^4$ cells/well) and incubated. After 1 day, virus was inoculated into cells at each multiplicity of infection (MOI): SARS-CoV-2$^{WK-521}$,0.33; SARS-CoV-2$^{QHN001}$(alpha), 25; SARS-CoV-2$^{TY8-612}$(beta), 25; SARS-CoV-2$^{TY7-501}$ (gamma), 20; SARS-CoV-2$^{KI734}$ (delta), 20; SARS-CoV-2$^{S356}$ (kapp), 20; SARS-CoV-2$^{00012}$(omicron BA.1), 32; SARS-CoV-2$^{02037}$ (omicron BA.2), 33; SARS-CoV-2$^{14631}$ (omicron BA.5), 30. Anti- SARS-CoV-2$^{WK-521}$assay using VeroE6 cells were performed with and without 2 μM P-glycoprotein inhibitor, CP-100356 (efflux inhibitor, EI) (Sigma-Aldrich, Co. LLC) addition. After an additional 3–4 days, cell culture supernatants were harvested and viral RNA was extracted using a QIAamp viral RNA minikit (Qiagen, Hilden, Germany), and quantitative RT-PCR (RT-qPCR) was then performed using One Step PrimeScript III RT-qPCR mix (TaKaRa Bio, Shiga, Japan) and a 7500 Fast Real-Time PCR Instrument (Applied Biosystems, Waltham, MA, USA) following the instructions of the manufacturers. The primers and probe used for detecting SARS-CoV-2 nucleocapsid (1) were 5'- AAATTTTGGGGAC-CAGGAAC-3' (forward), 5'- TGGCAGCTGTGTAGGTCAAC-3' (reverse), and 5'-FAM- ATGTCGCGCATTGGCATGGA-black hole quencher 1 (BHQ1)-3' (probe). To determine the cytotoxicity of each compound, cells were seeded in a 96-well plate ($2 \times 10^4$ cells/well). One day later, various concentrations of each compound were added, and cells were incubated for additional 3 days. The 50% cytotoxic concentrations (CC$_{50}$) values were determined using the WST-8 assay and Cell Counting Kit-8 (Dojindo, Kumamoto, Japan).

### Immunocytochemistry

Cells in a 96-well microtiter culture plate were fixed with 4% paraformal- dehyde–phosphate-buffered saline (PBS) for 15 min, washed with PBS (300 μl/well) three times for 5 min each time, and then blocked with a blocking buffer (10% goat serum, 1% bovine serum albumin [BSA], 0.3% Triton X-100, PBS 1x) for 1 h. After removal of the blocking buffer, the cells were immediately stained with a convalescent IgG fraction (concentration at 2.8 μg/mL), which was isolated from serum of a convalescent COVID-19 individual using a spin column-based antibody purification kit (Cosmo Bio, Tokyo, Japan) overnight at 4 °C. The stained cells were washed with PBS (300 μl/well) three times for 5 min each time, and the cells were incubated with secondary antibody goat polyclonal anti-human IgG-Alexa Fluor 488 Fab fragment antibody (concentration at 2.5 μg/mL) (catalog number:109-547-003) (Jackson ImmunoResearch Laboratories, Inc., West Grove, PA, USA), together with Texas Red-X dye-conjugated phalloidin (Thermo Fisher Scientific) for F-actin visualization for 2 h. After washing of the cells with PBS (300 μl/well) three times for 5 min each time, DAPI (4′,6-diamidino-2-phenylindole) solution (Thermo Fisher Scientific)–PBS (50 μl/well) was added to stain nuclei. Signals were acquired with a Cytation 5 cell imaging multi-mode reader (BioTek, Winooski, VT, USA).

### Determination of intracellular concentrations of compounds

The levels of intracellular concentrations of TKB245, TKB248, and Nirmatrelvir, which represent the balance between the penetration through the membrane and intracellular degradation of the compounds, were determined in VeroE6 and Hela-ACE2-TMPRSS2 cells. Three million cells were incubated with each drug (10, 50, and 100uM final concentration) at 37 °C for 6 h. Cells were harvested and washed with phosphate-buffered saline (PBS) three times, and cell pellets were resuspended in 70% methanol solution, and the suspensions were boiled at 95 °C for 5 min with shaking. The boiled suspensions were cooled to room temperature and centrifuged at $15,000 \times g$ for 10 min to separate cell debris from the solvent extract. Supernatants (solvent extracts) were transferred into new tubes, and the solvent was evaporated overnight. Dimethyl sulf-oxide (DMSO) (50 uL per tube) was added to the dried tubes, and the tubes were incubated at 37 °C for 1 h with shaking. Samples were then analyzed using TOF LC/MS. Drugs were separated on a VyDac C18 5 um-particle-size column (1.0 mm by 150 mm) using a gradient of solvent A (water–0.1% formic acid [FA]) and solvent B (acetonitrile–0.1% FA). The flow rate was set to 0.5 ml min-1, and the column was equilibrated with 95% solvent A and 5% solvent B. Following each injection, solvent B was increased to 55% over a 20 min period (2.5% increase per min). At 21 min solvent B was increased to 95% in 1 min and then returned to starting conditions over the next 1 min. Each drug was detected by TOF-MS using an Agilent 6230 mass spectrometer in selective ion monitoring mode (SIM). The sodium adducts of each drug provided the most prominent peak and therefore were used for detection purposes although the parent ions provided the relative results. The amount of drug obtained in the extracts was determined by comparison to the standards of each purified drug in DMSO. Each compound was confirmed by both elution time from the column and molecular weight by mass spectrometry. The above-mentioned LC/MS analysis was performed complying with the community requirements by Alseekh et al.[38]. and the detailed conditions are shown in source data.

### Animals

Jcl:ICR (ICR) female mice and PXB-mice (cDNA-uPA/SCID chimeric male mice with humanized liver with more than 70% of the liver replaced with human hepatocytes) were obtained from CLEA Japan (Tokyo, Japan) and PhoenixBio Co., Ltd.(Hiroshima, Japan), respectively. hACE2 knock-in mice, which were generated by inserting human ACE cDNA directly under the start codon in exon two of mouse Ace2 by the CRISPR/Cas9 system, were obtained from the RIKEN BioSource Center, Tukuba, Japan (official strain name: C.Cg-Ace2em1(ACE2)Okt: strain number RBRC11565). All mice were housed in an air-conditioned animal room at 23 ± 5 °C with a relative humidity of 55 ± 25°% under specific pathogen-free conditions, with a 12 h light/dark cycle (08:00 – 20:00/20:00 – 08:00). All mice were fed a standard rodent CRF-1 diet (Oriental yeast CO., LTD., JAPAN) and had ad libitum access to water. All animal experiments were approved by the President of NCGM, the University of Tokyo, and PhoenixBio Co., Ltd., following consideration by the Institutional Animal Care and Use Committee of NCGM (approval ID: no. 21057), the University of Tokyo (approval ID: PA19-72) and PhoenixBio Co., Ltd.(approval ID: 2722), and were carried out in accordance with institutional procedures, national guidelines and the relevant national laws on the protection of animals.

## Pharmacokinetic studies of TKB compounds in ICR mice and PXB-mice

For PK studies of TKB125 and TKB198, the ICR mice with body weights ranging from 20 to 25 g ($n$ = 2 or 3) were intravenously (i.v.) administered 2 mg/kg of 5 h, or TKB125. At various time points after administration (15, 30, 60, 120, and 240 min), blood samples were collected from the retro-orbital venous plexus under sevoflurane anesthesia and centrifuged at $3000 \times g$ for 15 min to obtain plasma. For PK studies of TKB245, TKB248, and nirmatelvir, the PXB mice with body weights ranging from 17 to 20 g ($n$ = 3 per each group) were i.v. administered 10 mg/kg of TKB245, TKB248, or nirmatelvir. The same mice were also perorally (p.o.) treated with 10 mg/kg of TKB245, TKB248, or nirmatelvir after a two-week washout. At various time points after administration (i.v., 0.167, 1, 4, 8, and 24 h; i.g., 0.167, 1, 4, 8, and 24 h), blood samples (~30 μL/time point) were collected from the retro-orbital venous plexus under sevoflurane anesthesia using heparinized syringes and centrifuged at $3000 \times g$ for 10 min to obtain plasma (~12 μL/time point). All the antiviral agents studied in the present work were solubilized in saline containing 5% DMSO and 9.5% cremophor EL (Sigma-Aldrich, Co. LCC).

## Antiviral efficacy against Delta, BA.1and BA.2 variants

A total of 13–14-week-old male hACE2 knock-in mice were used in this study. Baseline body weights were measured before infection. Under isoflurane anesthesia, mice were intranasally inoculated with $5 \times 10^5$ PFU/animal of Delta (hCoV-19/USA/WI-UW-5250/2021), $5 \times 10^5$ PFU/animal of BA.1 (hCoV-19/Japan/NC928-2N/2021) or $1 \times 10^5$ PFU/animal of BA.2 (hCoV-19/Japan/UT-NCD1288-2N/2022). TKB245 [100 mg/kg (in 600 μl, twice daily], TKB248 [120 mg/kg (in 600 μl, twice daily] or vehicle (in 600 μl) were administered intraperitoneally 2 h post infection in mice. TKB245 and TKB248 were solubilized in saline containing 5% DMSO and 9.5% cremophor EL. Body weight was monitored daily for 3 days. For virological and pathological examinations, the animals were euthanized at 2 and 3 dpi. The virus titers in the lungs were determined by plaque assays on VeroE6/TMPRSS2 cells. All animal experiments with SARS-CoV-2 were performed in animal biosafety level three (ABSL3) facility at the University of Tokyo.

## Statistical analyses of the changes in the viral loads in treated mice

We compared log-transformed virus titers against each SARS-CoV-2 variants between the vehicle and TKB245 using Wilcoxon rank-sum test. Similarly, comparisons between vehicle and TKB248 were also performed. The p values and mean viral titers were presented and mean ± standard deviation was plotted in the figure. We also compared viral titers between vehicle- and TKB245-/TKB248-treated mice using repeated ANOVA with days and mutant strains as fixed effects and individuals as a random effect.

## Protein expression and preparation

The SARS-CoV-2 $M^{pro}$-encoding sequence were cloned into pGEX-4T1 vector (Genscript) with N-terminal self-cleavage site (SAVLQ/SGFRK) and at the C-terminus, the construct codes for the human rhinovirus 3 C PreScission protease cleavage site (SGVTFQ↓GP) connected to a His6 tag. The plasmid constructs were transformed into BL21 Star™ (DE3) cells (Thermo Fisher Scientific). The cultures were grown in Terrific Broth media supplemented with ampicillin (Quality Biological, Gaithersburg, MD). Protein expression was induced by adding 1 mM iso-propyl beta-D-thiogalactopyranoside at an optical density of 0.8 at 600 nm and the cultures were maintained at 20 °C overnight. SARS-CoV2 $M^{pro}$ were purified first by affinity chromatography using TALON™ cobalt-based affinity Resin (Takara Bio). The authentic N-terminus is generated by 3CL $M^{pro}$ autoprocessing during expression, whereas the authentic

C-terminus is generated by the treatment with PreScission protease and the resulting authentic 306 amino acid $M^{pro}$ were further purified by SEC using a HiLoad Superdex 200 pg column (GE Healthcare) in 20 mM Tris, pH 7.5, 150 mM NaCl, and 2 mM DTT. Finally, the purified and concentrated SARS-CoV $M^{pro}$ (6.6–8.15 mg/mL) was stored in 200 mM ammonium acetate (pH 6.7).

## Native mass spectrometry

The authentic SARS-CoV-2 $M^{pro}$ was diluted to 7.5 μM in 10 mM ammonium acetate (pH 6.8) and the $M^{pro}$ was treated with either DMSO, TKB245, TKB248, or nirmatrelvir to give a final concentration of 15 μM for each compound. Glycine-added $M^{pro}$ was diluted to 10 μM and treated with either DMSO, TKB245, TKB248, or nirmatrelvir to give a final concentration of 50 μM for each compound. Each sample solution in native condition was incubated at room temperature and introduced to the ESI-QTOF mass spectrometer (impact II, Bruker Daltonics Bremen, Germany) through an infusion pump at a flow rate of 3 μL/min. Samples were ionized in positive ion mode with following ion source parameters: Dry heater: 200 ˚C, dry gas: 3.0 L/min, capillary voltage: 4500 V, end plate offset: −400 V, charging voltage: 2000V. MS scans have been acquired at a spectra rate of 1 Hz at a mass range from 500 to 6000 m/z. Molecular weights by protein deconvolution were determined using DataAnalysis 4.4 (Bruker Daltonics, Bremen, Germany). All species detected by native MS including their theoretical molecular weights, experimental molecular weights, and mass errors are listed in Supplementary Tables 3 and 4.

## LC-MS/MS analysis

A total of 20 μl of acetonitrile was added to 5 μl of plasma sample. The sample was stored for 15 min at 4 ˚C to achieve optimal protein precipitation. After centrifugation at $15{,}000 \times g$ at 4 ˚C, the obtained supernatant was added trifluoroacetic acid to give a final concentration of 0.05% for LC-MS/MS analysis. To detect the compounds, analysis was done using a quadrupole-time-of-flight (QTOF) mass spectrometer equipped with a Captive Spray electrospray ionization platform in the positive mode (impact II, Bruker Daltonics Bremen, Germany) with liquid chromatography (Ultimate 3000 HPLC, Thermo Fisher scientific). 1 μl of prepared sample was injected and concentrated on an Acclaim PepMap100 C18 trap column (Thermo Fisher scientific) at flow rate of 20 μl/min. For sample separation, Acclaim PepMap 100 C18LC column (0.075 mm × 150 mm, 2 μm particle) (Thermo Fisher scientific) was used in conditions of isocratic mode of 95% acetonitrile 0.1% FA for 7 min, flow rate of 300 nl/min and temperature of 35 ˚C. Compounds were approximately eluted at from 5 to 6 minutes. Following ion source parameters have been applied: Dry Heater: 150 ˚C, Dry Gas: 8.0 L/min, Capillary voltage: 1000 V, End plate offset: −500 V. Quantification was performed using multiple reaction monitoring (MRM). Precursor ion and quantifier ion in MRM transition of TKB245, TKB248 or nirmatrelvir were m/z654.2 and m/z110.1, m/z670.2 and m/z291.1 or m/z500.2 and m/z110.1, respectively. MS scans have been acquired at a spectra rate of 1 Hz at a mass range from 50 to 900 m/z. QuantAnalysis Ver2.2.1 (Bruker Daltonics, Bremen, Germany) was used for taking peak area of chromatogram of quantifier ions from MRM. The above-mentioned LC/MS analysis was performed complying with the community requirements by Alseekh et al[38]. and the detailed conditions are shown in source data.

## Crystallization of $M^{pro}$ and TKB245 or TKB248

The $M^{pro}$ was concentrated up to 3 mg/mL and incubated with 300 μM TKB245 or TKB248 for 1 h before crystallization. Crystals were grown using hanging drop vapour diffusion method at 20 °C. The reservoir solution for TKB245 and nirmatrelvir contained 0.1 M MES pH 6.8, 15% polyethylene glycol (PEG) 6000 and 3% DMSO or 0.1 M MES pH 6.8, 15% PEG 6000 and 3% DMSO and the compounds were crystalized at

room temperature, while that solution for TKB248 contained 0.1 M MES pH 5.6, 10% PEG 6000, and 3% DMSO and it was crystalized at 4 °C. Crystals were soaked briefly in a cryoprotection solution containing 0.1 M MES pH 6.0, 35% PEG 400 5% DMSO. X-ray data were collected at SPring-8 BL41XU (Hyōgo, Japan) and processed using DIALS using xia2 incorporated in CCP4i2[39]. The source wavelength for the data collection was 1.0 Å. Data collection statistics are shown in Supplementary Table S3. The phase problem was solved by molecular replacement using MolRep[40] using the 1.25 Å structure of M$^{pro}$ (PDB ID: 7JKV) as a model. All water molecules and ligand atoms were omitted from the starting model. Subsequent cycles of refinement were performed in REFMAC5[41]. Structure file of TKB245 and TKB248 were generated using the Dundee PRODRG2 server[42] and manually fitted to the electron density. All structural figures were produced with PyMOL[43] and UCSF Chimera[44].

## Reporting summary

Further information on research design is available in the Nature Portfolio Reporting Summary linked to this article.

## Data availability

All data supporting the findings in this study are available from the source data file provided in this paper and the corresponding author upon request. Crystal structure data that support the findings of this study have been deposited in Protein Data Bank with the PDB IDs: 8DOX [https://doi.org/10.2210/pdb8DOX/pdb] and 8DPR [https://doi.org/10.2210/pdb8DPR/pdb]. Previously published crystal structures used in this study are available under the PDB IDs 7JKV[https://doi.org/10.2210/pdb7JKV/pdb]) and 7RFW [https://doi.org/10.2210/pdb7RFW/pdb]). Source data are provided with this paper.

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

## Acknowledgements

The present work was supported by a grant for Development of Novel Drugs for Treating COVID-19 (H.M., 19A3001) (S.H., 20A2001D) from the Intramural Research Program of National Center for Global Health and Medicine (H.M.), in part by the Intramural Research Program of the Center for Cancer Research, National Cancer Institute, National Institutes of Health (H.M.) in part by the Japan Agency for Medical Research and Development (AMED) (grant numbers JP20fk0108257, and JP20fk0108510 to H.M.; JP21fk0108480 to T.S.; JP22wm0125002 and JP223fa627001 to Y.K.). The synchrotron radiation experiment was performed at BL41XU of SPring-8 with the approval of the Japan Synchrotron Radiation Research Institute (JASRI) (proposal no. 2021A2725, 2021B2560). This study utilized the high-performance computational capabilities of the Biowulf Linux cluster at the National Institutes of Health, Bethesda, MD (https://hpc.nih.gov). We also acknowledge support from the Biomedical Research Core of the Tohoku University Graduate School of Medicine. The authors thank Ms. Asuka Fujiwara and Ms. Mariko Kato, National Center for Global Health and Medicine, and Ms. Yuko Sato, National Institute of Infectious Diseases, for technical help. We gratefully acknowledge all data contributors, i.e., the Authors and their Originating laboratories responsible for obtaining the specimens, and their Submitting laboratories for generating the genetic sequence and metadata and sharing via the GISAID Initiative, on which this research is based. The content of this publication does not necessarily reflect the views or policies of the Department of Health and Human Services, nor does mention of trade names, commercial products, or organizations imply endorsement by the U.S. Government.

## Author contributions

Conceptualization, N.H.-K. and H.M.; Investigation, N.H.-K., K.T., H.H., H.B., M.K., M.I., H.O.-A., T.I., T.K., K.N., Yukiko S., N.T., N.K., S.H., D.D., Y.U., Yosuke S., S.S., A.N., S.J., Yoshikazu S., Y.T., K.T., K.M., K.Y., S.I., S.O., T.S., T.O., S.M., Y.K., H.T., and H.M.; Data curation, N.H.-K. and H.M.; Original draft writing, N.H.-K. and H.M.; Writing-review and editing, all authors; Project Administration, H.M.; Funding acquisition, T.S., Y.K., and H.M.

## Competing interests

The authors declare no competing interests.

## Additional information

[1]Department of Refractory Viral Diseases, National Center for Global Health and Medicine Research Institute, Tokyo, Japan. [2]Department of Medicinal Chemistry, Institute of Biomaterials and Bioengineering, Tokyo Medical and Dental University, Tokyo, Japan. [3]Department of Infectious Diseases, International Research Institute of Disaster Science, Tohoku University, Miyagi, Japan. [4]Experimental Retrovirology Section, HIV and AIDS Malignancy Branch, National Cancer Institute, NIH, Bethesda, MD, USA. [5]Division of Virology, Institute of Medical Science, University of Tokyo, Tokyo, Japan. [6]The Research Center for Global Viral Diseases, National Center for Global Health and Medicine Research Institute, Tokyo, Japan. [7]Department of Laboratory Animal Medicine, Research Institute, National Center for Global Health and Medicine, Tokyo, Japan. [8]Department of Environmental and Molecular Health Sciences, Faculty of Life

Sciences, Kumamoto University, Kumamoto, Japan. [9]Center for Clinical Sciences, National Center for Global Health and Medicine, Tokyo, Japan. [10]Structural Biology Division, Japan Synchrotron Radiation Research Institute, Hyogo, Japan. [11]Department of Infectious Diseases, Tohoku University Graduate School of Medicine, Miyagi, Japan. [12]Division of Pharmacology and Therapeutics, Graduate School of Pharmaceutical Sciences, Kumamoto University, Kumamoto, Japan. [13]AIDS Clinical Center, National Center for Global Health and Medicine, Tokyo, Japan. [14]Tokyo Metropolitan Institute of Public Health, Tokyo, Japan. [15]Department of Pathology, National Institute of Infectious Diseases, Tokyo, Japan. [16]Influenza Research Institute, Department of Pathobiological Sciences, School of Veterinary Medicine, University of Wisconsin-Madison, Madison, WI, USA. [17]Kumamoto University Hospital, Kumamoto, Japan.
✉e-mail: hmitsuya@hosp.ncgm.go.jp

