## [Peer Review File · Nature Communications]

REVIEWER COMMENTS

Reviewer #1 (Remarks to the Author):

Based on the benzothiazole-containing Mpro inhibitor, 5h, a series of analogs were designed and synthesized. Two e 4-fluoro-benzothiazole-containing compounds, TKB245 and TKB248, were found to have enhanced antiviral activity and pharmacokinetic properties. Specifically, these two compounds contain 6,6-dimethyl-3-azabicyclohexane and trifluoroacetyl at P2 and P4 substitutions, respectively. These two compounds had comparable enzymatic inhibition against Mpro compared with nirmatrelvir, and had improved antiviral activity in cell culture. Both compounds also showed broad-spectrum antiviral activity against SARS-CoV-2 variants in several cell lines. In vivo antiviral efficacy study showed that both compounds were able to reduce the viral titer in the lung in hACE2 mice infected with SARS-CoV-2. It was claimed that both compounds might be superior to nirmatrelvir. Native mass spectrometry experiment was used to characterize the mechanism of action. TKB245 but not TKB248 were found to shift the Mpro monomer dimer equilibrium to dimer. X-ray crystal structures were also solved, which is consistent with the predicted binding mode.

Comments are:

1. The authors should be cautious in claiming that TKB245 and 248 are better than nirmatrelvir. In this study, both compounds were dosed by ip injection, and nirmatrelvir was not included as a control for head-to-head comparison. The major advantage of nirmatrelvir is the oral administration. The authors should comment why these compounds were not dosed by oral gavage.
2. Fig 2A, there is almost no difference in viral titer at Day 3 for TKB245 compared to control. Any explanations? Similar results were shown in Fig. 2B for compound TKB248 against Omicron BA2 strain at Day 2 and Day 3.
3. "Moreover, the emergence of SARS-CoV-2-resistant variants against these first generation drugs have raised significant concerns¹⁰"

Comment: reference 10 is not the correct citation for Mpro resistance as it did not have experimental validation. Instead, the following references should be considered.

<https://doi.org/10.1101/2022.08.07.499047>

<https://doi.org/10.1101/2022.06.28.497978>

4. TKB245 and nirmatrelvir bind to Mpro and promote Mpro dimerization.

Comment: the authors should be cautious in claiming that drug binding induces dimerization. The conclusion was based on native mass spect experiments, and the conditions do not reflect the physiological condition. Nirmatrelvir and similar compounds are not known to induce dimerization.

5. What is the drug formulation of drugs used in the in vivo PK and antiviral efficacy study?
6. Please double check the chemical structures of TKB245 and 248. They are identical.

7. The in vivo PK results were not shown.

8. For the in vivo antiviral efficacy study, what about the body weight change and survival rate? It is odd that the authors only showed the results of viral titer in the lung. How about the viral titers in other tissue and organs?

9. How about the target specificity of these compounds against host cysteine proteases such as cathepsin L, calpains and others?

Reviewer #2 (Remarks to the Author):

The manuscript describes the synthesis and characterization of a new series of Mpro inhibitors. The activity data were promising and the best compound appeared to be more potent than nirmatrelvir. But the investigation of the mechanism will need to be improved.

1) The abstract and the text seems to suggest there is some difference between the inhibition mechanisms of TKB245 and TKB248, i.e., TKB245 promotes dimerization and TKB248 does not. However, the differences in their observations in Fig. 3 could simply have been due to the weaker affinity of TKB248. There is a ten fold difference between the IC50's of the two compounds (7 uM vs 74 uM). The monomer/dimer analysis was done at a compound concentration of 15 uM. The pyrrolidone ring of both compounds would stabilize the S1 site which is right next to the dimer interface. So both compounds should be able to promote the dimer formation. Have the authors tried higher concentrations of TKB248 in these experiments?

2) There is only one atom difference (oxygen vs sulfur) between TKB245 and TKB248. Did the crystal structures provide any information for what caused the differences in the IC50's?

3) TKB245 is more potent than nirmatrelvir and the authors partly attributed this to the TKB245 benzothiazole side chain that interacts with the S1' site. This is reasonable as nirmatrelvir does not form contacts with this site. But the two compounds (TKB245 vs nirmatrelvir) also have different warheads to react with C145. Can the authors comment on the reactivity of these two warheads relative to one another?

4) Table 1, the chemical structures of TKB245 and TKB248 are the same.

5) Supplemental Table 2, the 'resolution' row does not have the information for the highest resolution shell, only the lowest and highest resolution of the full range of data.

6) Line 543, the citation for Chimera is missing.

Reviewer #3 (Remarks to the Author):

*The investigative team presented a highly important set of observations regarding the characterization of potent SARS-CoV-2 Mpro inhibitors with a fluorinated benzothiazole moiety.

*The presented work is of significance to the field by advancing the development of more potent coronavirus protease inhibitors.

*The research team uses a diverse set of experimental techniques. However, some of the figures and figure legends look blurry (Figures 1, 2, 3, and 4). The figures in the current form are not suitable for publication and should be replaced by higher resolution figures.

*The analysis of the data is appropriate, and the main conclusions are well supported. There is one exception regarding the analysis of the mass spectrometry experiment of the Mpro-inhibitor interaction. Additional biophysical and biochemical experiments are needed to be able to conclude that "Native mass spectrometry analysis revealed that predominantly two TKB245 molecules bind to dimer Mpro, promoting dimerization, while mostly on TKB248 molecule binds to the enzyme, failing to promote dimerization." This statement is repeated both in the Abstract and the Discussion. Detailed mechanistic studies are needed to demonstrate the role of these relatively small molecules in dimerization of the enzyme.

*The experiments are described in appropriate detail.

*In summary, the research presented is solid with the exception of dimerization of Mpro promoted by inhibitor binding.

*For publication of the work, the figures should be replaced with high resolution copies and the interpretation of the mass spectrometry experiment should be revised.

Center for Cancer Research
HIV & AIDS Malignancy Branch
The Experimental
Retrovirology Section

9000 Rockville Pike
Building 10, Room 5A11
Bethesda, Maryland 20892
(301) 496-9238; 9239
(301) 402-3631
(301) 402-0709 FAX

U.S. Department of Health
and Human Services
Public Health Service
National Institutes of Health

December 12, 2022

Re: Revised Manuscript # NCOMMS-22-37236

“Identification of SARS-CoV-2 M^{pro} Inhibitors containing P1’ 4-fluorobenzothiazole Moiety Highly Active against SARS-CoV-2” for publication in *Nature Communications*

To the Reviewers:

Thank you very much for your constructive advices and very helpful suggestions, which helped us very much improve and maximize our paper. As below, we have described all the changes we made as point-by-point response to your advices/suggestions.

Reviewer 1:

Comments/Suggestions #1:

The authors should be cautious in claiming that TKB245 and 248 are better than nirmatrelvir. In this study, both compounds were dosed by ip injection, and nirmatrelvir was not included as a control for head-to-head comparison. The major advantage of nirmatrelvir is the oral administration. The authors should comment why these compounds were not dosed by oral gavage.

Our Response:

As has been documented in the original supplementary Figure 1 (supplementary Figure 2 in the revised MS), (Supple.Fig2a) as perorally administered (in the absence of ritonavir in human liver-chimeric PXB-mice), the PK profile of TKB248 was the best, followed by TKB245, and then nirmatrelvir; (Supple.Fig2b) as peroral and intravenous administrations were compared for TKB245, the PK of peroral administration was better than that of intravenous administration as examined in PXB mice; and (Supple.Fig2d) as intraperitoneal administration was compared to peroral TKB245 administration, the PK profile of *i.p.* administration was better as examined in ICR mice. Thus, intraperitoneal TKB245 administration was chosen. Another reason of the choice of *i.p.* administration was in part that the less labor and safety issue for the lab workers were considered favorable. The reason of the use of PXB-mice was that such mice carry human hepatocytes and their PK profiles are closer to PKs in humans.

In the revised version of the manuscript, we took cautions not to mention “TKB245 and TKB248 are better than nirmatrelvir”. Instead, we have now precisely described that “TKB245 and TKB248 are more potent than nirmatrelvir, molnupiravir, and ensitrelvir in cell-based assays using various target cells”.

Comments/Suggestions #2:

Fig 2A, there is almost no difference in viral titer at Day 3 for TKB245 compared to control. Any explanations? Similar results were shown in Fig. 2B for compound TKB248 against Omicron BA2 strain at Day 2 and Day 3.

Our Response:

The concern Reviewer 1 has is truly legitimate. Due to the paucity of the compounds, TKB245 and TKB248, and the available mice for the *in vivo* anti-SARS-CoV-2 activity of the two compounds, we decided to use five mice per group in the experiments in question.

When we completed the experiments, a variability was seen in the data obtained. As seen in Figure 2, since the sample size for this study was rather small (n=5), the results of subgroup analysis for mutant strains (SARS-CoV-2_{NC928-2N}^{Omicron_BA.1} and SARS-CoV-2_{UW-5250}^{Delta}) and days (day 2 and day 3) exhibited a substantial variability, and there were subgroups that showed no effects for TKB245's anti-SARS-CoV-2 activity compared to the no-drug control. On the other hand, using repeated ANOVA with days and mutant strains as fixed effects and individuals as a random effect, TKB245 showed significantly lower virus titer compared to the no-drug control [-1.34 (95%CI: -1.85 , -0.82)]. Similarly, TKB248 showed significantly lower virus titers compared to no-drug controls [-1.01 (95%CI: -1.41, -0.60)].

We have mentioned these points and limitations **in the Results section of the revised version of the manuscript (Pages 8-9, Lines 211-216).**

Comments/Suggestions #3:

“Moreover, the emergence of SARS-CoV-2-resistant variants against these first generation drugs have raised significant concerns¹⁰”

Comment: reference 10 is not the correct citation for Mpro resistance as it did not have experimental validation. Instead, the following references should be considered.
<https://doi.org/10.1101/2022.08.07.499047>
<https://doi.org/10.1101/2022.06.28.497978>

Our Response:

At the time of writing our original version of the MS, we found only one reference by Motyan *et al.* (*Int. J Mol.Sci* 23:3507 doi: 10.3390/ijms23073507). In the revised version of the manuscript, we have updated the references regarding the emergence of SARS-CoV-2-resistance variants and cited two publications by Iketani *et al.* (*Nature* 2022) and Hu *et al.* (*bioRxiv* 2022) as Reviewer 1 suggested **in the References section of the revised version of the manuscript (Pages 23-24, Lines 614-622).**

Comments/Suggestions #4:

TKB245 and nirmatrelvir bind to Mpro and promote Mpro dimerization.

Comment: the authors should be cautious in claiming that drug binding induces dimerization. The conclusion was based on native mass spect experiments, and the conditions do not reflect the physiological condition. Nirmatrelvir and similar compounds are not known to induce dimerization.

Our Response:

Comments by Reviewer 1 have been taken seriously. He/she is right that the data by native mass spectrometry do not immediately mean that the same M^{pro} dimerization events occur under the physiologic conditions (in the cytoplasm where a number of cellular and viral proteins and salts *etc.* exist). However, it is clear that under the conditions we used in the native mass spectrometry, the amounts of M^{pro} protomer significantly decreased in the presence of TKB245, TKB248 (at 50 μM in the newly generated Supplementary Figure 6), and nirmatrelvir. The data together with the original Figure

3 and Supplementary Figure 6 strongly suggest that all the three M^{Pro} inhibitors similarly promote the dimerization of two protomers. Taking the comments by Reviewer 1, we have softened the original wording “promoting M^{Pro} dimerization” to “apparently promoting M^{Pro} dimerization” in both of the abstract (Page 3, Lines 61-62) and text of the revised MS (Page 9, Lines 229-234).

Comments/Suggestions #5:

What is the drug formulation of drugs used in the in vivo PK and antiviral efficacy study?

Our Response:

All the antiviral agents studied in the present work were solubilized in saline containing 5% DMSO and 9.5% cremophor EL. We described this information in the **Materials and Methods section of the revised version of the manuscript (Page 18, Lines 469-471)**.

Comments/Suggestions #6:

Please double check the chemical structures of TKB245 and 248. They are identical.

Our Response:

We are terribly sorry that it was our inadvertent mistake. We have now put the correct structure of TKB245 in the revised Table 1.

Comments/Suggestions #7:

The in vivo PK results were not shown.

Our Response:

We conducted the PK study of TKB198, TKB245, TKB248, as well as nirmatrelvir. We have mentioned the PK profiles of those antiviral agents in the original Table 1 and the original Supplementary Figure 1. In particular, we have mentioned the detailed PK profiles of TKB245, TKB248, and nirmatrelvir in the original Supplementary Figure 2.

Comments/Suggestions #8:

For the in vivo antiviral efficacy study, what about the body weight change and survival rate? It is odd that the authors only showed the results of viral titer in the lung. How about the viral titers in other tissue and organs?

Our Response:

Although the hACE2-knocked-in mice used in this study are susceptible to viral infection and support viral replication in the lung like in the hamster model we have previously published (Imai *et al. PNAS* 117:16587–16595, 2020), the virus does not significantly replicate in other parts of the body. Thus, the data of the viral titers were omitted except those in lungs. The hACE2-knocked-in mice did not succumb to death at all and the survival rate was 100%. Regarding the possible body weight changes, we monitored the weights throughout the study periods. We have added the body weight change and survival data in a newly added Supplementary Figure 3 and described this information in the **Results section of the revised version of the manuscript (Page 8, Lines 203-206)**.

Comments/Suggestions #9:

How about the target specificity of these compounds against host cysteine proteases such as cathepsin L, calpains and others?

Our Response:

In response to Reviewer 1’s query, we have newly determined the sensitivity of cathepsin L and calpain to TKB245, TKB248, and nirmatrelvir. As seen in the newly added Supplementary Figure 1, neither

of the human enzymes were significantly affected by any of the three M^{Pro} inhibitors. We have added these new data in the Results section of the revised version of the MS (Page 5, Lines 119-121).

Reviewer 2

Comments/Suggestions #1:

The abstract and the text seems to suggest there is some difference between the inhibition mechanisms of TKB245 and TKB248, i.e., TKB245 promotes dimerization and TKB248 does not. However, the differences in their observations in Fig. 3 could simply have been due to the weaker affinity of TKB248. There is a ten fold difference between the IC₅₀'s of the two compounds (7 μ M vs 74 μ M). The monomer/dimer analysis was done at a compound concentration of 15 μ M. The pyrrolidone ring of both compounds would stabilize the S1 site which is right next to the dimer interface. So both compounds should be able to promote the dimer formation. Have the authors tried higher concentrations of TKB248 in these experiments?

Our Response:

We appreciate the very reasonable comments by Reviewer 2 and we have now newly conducted additional experiments using a much higher concentration of TKB248, 50 μ M. As can be seen in newly added Supplementary Figure 6, the addition of 50 μ M TKB248 significantly increased the amount of M^{Pro} bound by two TKB248 molecules, followed by that of M^{Pro} bound by one molecule. There was only a tiny amount of unbound but dimerized M^{Pro} in the same chart. These new data strongly suggest that as Reviewer 2 suspected, TKB248 also promotes the dimerization process of two protomers. We have now mentioned this new observation that both TKB245 and TKB248 do bind to M^{Pro} and promote its dimerization in both of the abstract (Page 3, Lines 61-62) and text of the revised MS (Page 9, Lines 229-234).

Comments/Suggestions #2:

There is only one atom difference (oxygen vs sulfur) between TKB245 and TKB248. Did the crystal structures provide any information for what caused the differences in the IC₅₀'s?

Our Response:

The structural conformation of TKB245 or TKB248 complexed with M^{Pro} obtained by our X-ray crystallography showed no significant difference between the complexes. In the proximity of the two atoms, the oxygen and sulfur, there were no differences. The mechanism of the difference in the IC₅₀ value between TKB245 and TKB248 is not clear at this time.

Comments/Suggestions #3:

TKB245 is more potent than nirmatrelvir and the authors partly attributed this to the TKB245 benzothiazole side chain that interacts with the S1' site. This is reasonable as nirmatrelvir does not form contacts with this site. But the two compounds (TKB245 vs nirmatrelvir) also have different warheads to react with C145. Can the authors comment on the reactivity of these two warheads relative to one another?

Our Response:

Actually, understanding the contribution of warheads to anti-SARS-CoV-2 properties is an interesting aspect and we thank Reviewer 2 for rising this point. In this study, however, we have not designed compounds to directly address this question. It is rather difficult to precisely compare the reactivity of the two warheads, CN and CO, at this time. We are designing new nirmatrelvir derivatives based

on aldehyde warheads that could be used in direct comparison to understand the contribution of each warhead (aldehyde *versus* nitrile), but this will be examined in the future project. We have mentioned these points in the Discussion section in the revised version of the MS (Page 13, Lines 321-325).

Comments/Suggestions #4:

Table 1, the chemical structures of TKB245 and TKB248 are the same.

Our Response:

We are terribly sorry that it was our inadvertent mistake. We have now put the correct structure of TKB245 in the revised Table 1.

Comments/Suggestions #5:

Supplemental Table 2, the 'resolution' row does not have the information for the highest resolution shell, only the lowest and highest resolution of the full range of data.

Our Response:

We have added the exact values of the resolution of the full range of data in the revised MS.

Comments/Suggestions #6:

Line 543, the citation for Chimera is missing.

Our Response:

We are sorry that the citation for “Chimera” had been omitted. We have now added the correct citation for “Chimer” in the revised MS as follows (Page 22, Line 566):

Pettersen EF, Goddard TD, Huanwg CC, Couch GS, Greenblatt DM, Meng EC, Ferrin TE. UCSF Chimera – a visualization system for exploratory research and analysis. *J Comput Chem.* 2004 Oct;25(13):1605-12.

Reviewer 3

Comments/Suggestions #1:

The investigative team presented a highly important set of observations regarding the characterization of potent SARS-CoV-2 Mpro inhibitors with a fluorinated benzothiazole moiety.

Our Response:

We appreciate the kind comment Reviewer #3 made.

Comments/Suggestions #2:

The presented work is of significance to the field by advancing the development of more potent coronavirus protease inhibitors.

Our Response:

We again do appreciate the kind comment Reviewer #3 made.

Comments/Suggestions #3:

The research team uses a diverse set of experimental techniques. However, some of the figures and figure legends look blurry (Figures 1, 2, 3, and 4). The figures in the current form are not suitable for publication and should be replaced by higher resolution figures.

Our Response:

We are sorry that in the first phase of submission of manuscript in the Published does not allow us to send high resolution figures to the Reviewers (We have attempted to send the high resolution figures

in the first submission). We believe that *Nat Commun* will use much greater resolution figures when the manuscripts is accepted and figures are printed.

Comments/Suggestions #4:

The analysis of the data is appropriate, and the main conclusions are well supported. There is one exception regarding the analysis of the mass spectrometry experiment of the Mpro-inhibitor interaction. Additional biophysical and biochemical experiments are needed to be able to conclude that "Native mass spectrometry analysis revealed that predominantly two TKB245 molecules bind to dimer Mpro, promoting dimerization, while mostly on TKB248 molecule binds to the enzyme, failing to promote dimerization." This statement is repeated both in the Abstract and the Discussion. Detailed mechanistic studies are needed to demonstrate the role of these relatively small molecules in dimerization of the enzyme.

Our Response:

Reviewer 1 gave the same criticism to the paper. We have now newly conducted additional experiments using a much higher concentration of TKB248, 50 μ M. As can be seen in newly added Supplementary Figure 6, the addition of 50 μ M TKB248 significantly increased the amount of M^{pro} bound by two TKB248 molecules, followed by that of M^{pro} bound by one molecule. There was only a tiny amount of unbound but dimerized M^{pro} in the same chart. These new data strongly suggest that as Reviewer 2 suspected, TKB248 also promotes the dimerization process of two protomers. We have now mentioned this new observation that both TKB245 and TKB248 do bind to M^{pro} and promote its dimerization in both of the abstract (Page 3, Lines 61-62) and text (Page 9, Lines 229-234) of the revised MS.

Comments/Suggestions #5:

The experiments are described in appropriate detail.

Our Response:

We appreciate the kind comment Reviewer #3 made.

Comments/Suggestions #6:

In summary, the research presented is solid with the exception of dimerization of M^{pro} promoted by inhibitor binding.

Our Response:

We do appreciate the points by Reviewer #3. We have conducted additional experiments and we now believe that we have solved the dimerization issues in the revised version of the MS.

Comments/Suggestions #7:

For publication of the work, the figures should be replaced with high resolution copies and the interpretation of the mass spectrometry experiment should be revised.

Our Response:

We are sorry that in the first phase of submission of manuscript in the Published does not allow us to send high resolution figures to the Reviewers (We have attempted to send the high resolution figures in the first submission). We believe that *Nat Commun* will use much greater resolution figures when the manuscripts is accepted and figures are printed. As mentioned above (in response to #4 query), we have conducted additional mass spectrometry experiments and we now believe that we have solved the dimerization issues in the revised version of the MS (Page 9, Lines 229-234).

We believe that the revised version of the manuscript has now been very much strengthened and clarified as the consequence of the careful and constructive process of the review. We sincerely hope that you might now find the revised version of the manuscript acceptable for publication in *Nature Communications*.

Sincerely yours,

Hiroaki Mitsuya, M.D., Ph.D.
Chief & Principal Investigator, Experimental Retrovirology Section
National Cancer Institute, NIH, Bethesda, MD, USA
Director
National Center for Global Health & Medicine Research Institute
Shinjuku, Tokyo, Japan

Nobuyo Higashi-Kuwata, D.V.M., Ph.D.
National Center for Global Health & Medicine Research Institute
Shinjuku, Tokyo, Japan

REVIEWERS' COMMENTS

Reviewer #1 (Remarks to the Author):

Comments from the previous round of review were properly addressed. I therefore recommend acceptance.

Reviewer #2 (Remarks to the Author):

The authors have performed additional experiments and revised the manuscript based on the previous reviews. In particular, the new experiments now show that both TKB245 and TKB248 can stabilize the Mpro dimer. The revised discussion is also more balanced. These results can provide useful information for the on-going efforts to develop COVID-19 therapeutics.

Reviewer #3 (Remarks to the Author):

I have carefully read the revised manuscript and the responses to all three reviewers' comments. I am satisfied with the authors' edits to address all criticisms and recommend the manuscript for publication.